# Using Statistics to Automate Stochastic Optimization

**Hunter Lang**      **Pengchuan Zhang**      **Lin Xiao**

Microsoft Research AI
Redmond, WA 98052, USA
{hunter.lang, penzhan, lin.xiao}@microsoft.com

## Abstract

Despite the development of numerous adaptive optimizers, tuning the learning rate of stochastic gradient methods remains a major roadblock to obtaining good practical performance in machine learning. Rather than changing the learning rate at each iteration, we propose an approach that automates the most common hand-tuning heuristic: use a constant learning rate until "progress stops", then drop. We design an explicit statistical test that determines when the dynamics of stochastic gradient descent reach a stationary distribution. This test can be performed easily during training, and when it fires, we decrease the learning rate by a constant multiplicative factor. Our experiments on several deep learning tasks demonstrate that this statistical adaptive stochastic approximation (SASA) method can automatically find good learning rate schedules and match the performance of hand-tuned methods using default settings of its parameters. The statistical testing helps to control the variance of this procedure and improves its robustness.

## 1   Introduction

Stochastic approximation methods, including stochastic gradient descent (SGD) and its many variants, serve as the workhorses of machine learning with big data. Many tasks in machine learning can be formulated as the stochastic optimization problem:

$$\text{minimize}_{x \in \mathbf{R}^n} \ \ F(x) \triangleq \mathbf{E}_\xi \big[ f(x, \xi) \big],$$

where $\xi$ is a random variable representing data sampled from some (unknown) probability distribution, $x \in \mathbf{R}^n$ represents the parameters of the model (e.g., the weight matrices in a neural network), and $f$ is a loss function. In this paper, we focus on the following variant of SGD with *momentum*,

$$\begin{aligned} d^{k+1} &= (1 - \beta_k)g^k + \beta_k d^k, \\ x^{k+1} &= x^k - \alpha_k d^{k+1}, \end{aligned} \qquad (1)$$

where $g^k = \nabla_x f(x^k, \xi^k)$ is a stochastic gradient, $\alpha_k > 0$ is the learning rate, and $\beta_k \in [0, 1)$ is the momentum coefficient. This approach can be viewed as an extension of the heavy-ball method (Polyak, 1964) to the stochastic setting.[1] To distinguish it from the classical SGD, we refer to the method (1) as SGM (Stochastic Gradient with Momentum).

Theoretical conditions on the convergence of stochastic approximation methods are well established (see, e.g., Wasan, 1969; Kushner and Yin, 2003, and references therein). Unfortunately, these asymptotic conditions are insufficient in practice. For example, the classical rule $\alpha_k = a/(k + b)^c$ where $a, b > 0$ and $1/2 < c \leq 1$, often gives poor performance even when $a$, $b$, and $c$ are hand-tuned. Additionally, despite the advent of numerous adaptive variants of SGD and SGM (e.g., Duchi et al., 2011; Tieleman and Hinton, 2012; Kingma and Ba, 2014, and other variants), achieving good performance in practice often still requires considerable hand-tuning (Wilson et al., 2017).

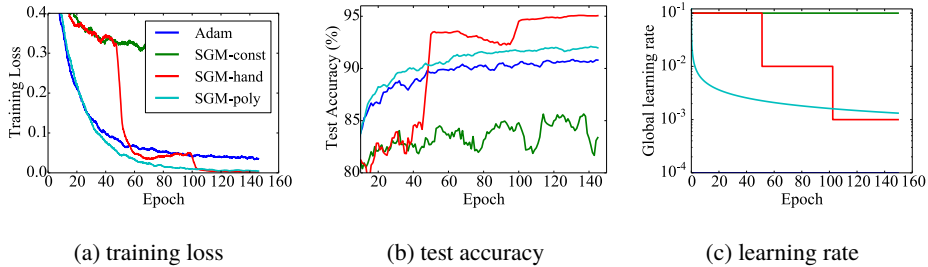

|                     |                   |                      |
| :-----------------: | :---------------: | :------------------: |
| (a) training loss   | (b) test accuracy | (c) learning rate    |

Figure 1: Smoothed training loss, test accuracy, and (global) learning rate schedule for an 18-layer ResNet model (He et al., 2016) trained on the CIFAR-10 dataset using four different methods (with constant momentum $\beta = 0.9$). Adam: $\alpha = 0.0001$; SGM-const: $\alpha = 1.0$; SGM-poly: $a = 1.0, b = 1, c = 0.5$; SGM-hand: $\alpha = 1.0$, drop by 10 every 50 epochs.

Figure 1 shows the training loss and test accuracy of a typical deep learning task using four different methods: SGM with constant step size (SGM-const), SGM with diminishing $O(1/k)$ step size (SGM-poly), Adam (Kingma and Ba, 2014), and hand-tuned SGM with learning rate scheduling (SGM-hand). For the last method, we decrease the step size by a multiplicative factor after a suitably long number of epochs ("constant-and-cut"). The relative performance depicted in Figure 1 is typical of many tasks in deep learning. In particular, SGM with a large momentum and a constant-and-cut step-size schedule often achieves the best performance. Many former and current state-of-the-art results use constant-and-cut schedules during training, such as those in image classification (Huang et al., 2018), object detection (Szegedy et al., 2015), machine translation (Gehring et al., 2017), and speech

---

**Algorithm 1:** General SASA method

**Input:** $\{x^0, \alpha_0, M, \beta, \zeta\}$
1 **for** $j \in \{0, 1, \ldots\}$ **do**
2      **for** $k \in \{jM, \ldots, (j+1)M - 1\}$ **do**
3          Sample $\xi^k$.
4          Compute $g^k = \nabla_x f(x^k, \xi^k)$.
5          $d^{k+1} = (1 - \beta)g^k + \beta d^k$
6          $x^{k+1} = x^k - \alpha d^{k+1}$
7          // collect statistics
8      **end**
9      **if** *test(statistics)* **then**
10          $\alpha \leftarrow \zeta \alpha$
11          // reset statistics
12      **end**
13 **end**

---

recognition (Amodei et al., 2016). Additionally, some recent theoretical evidence indicates that in some (strongly convex) scenarios, the constant-and-cut scheme has better finite-time last-iterate convergence performance than other methods (Ge et al., 2019).

Inspired by the success of the "constant-and-cut" scheduling approach, we develop an algorithm that can automatically decide when to drop $\alpha$. Most common heuristics try to identify when "training progress has stalled." We formalize stalled progress as when the SGM dynamics in (1), with constant values of $\alpha$ and $\beta$, reach a stationary distribution. The existence of such a distribution seems to match well with many empirical results (e.g., Figure 1), though it may not exist in general. Since SGM generates a rich set of information as it runs (i.e. $\{x^0, g^0, \ldots, x^k, g^k\}$), a natural approach is to collect some statistics from this information and perform certain tests on them to decide whether the process (1) has reached a stationary distribution. We call this general method SASA: statistical adaptive stochastic approximation.

Algorithm 1 summarizes the general SASA method. It performs the SGM updates (1) in phases of $M$ iterations, in each iteration potentially computing some additional statistics. After $M$ iterations are complete, the algorithm performs a statistical test to decide whether to drop the learning rate by a factor $\zeta < 1$. Dropping $\alpha$ after a fixed number of epochs and dropping $\alpha$ based on the loss of a held-out validation set correspond to heuristic versions of Algorithm 1. In the rest of this work, we detail how to perform the "test" procedure and evaluate SASA on a wide range of deep learning tasks.

## 1.1 Related Work and Contributions

The idea of using statistical testing to augment stochastic optimization methods goes back at least to Pflug (1983), who derived a stationary condition for the dynamics of SGD on quadratic functions

and designed a heuristic test to determine when the dynamics had reached stationary. He used this test to schedule a fixed-factor learning rate drop. Chee and Toulis (2018) recently re-investigated Pflug's method for general convex functions. Pflug's stationary condition relies heavily on a quadratic approximation to $F$ and limiting noise assumptions, as do several other recent works that derive a stationary condition (e.g., Mandt et al., 2017; Chee and Toulis, 2018). Additionally, Pflug's test assumes no correlation exists between consecutive samples in the optimization trajectory. Neither is true in practice, which we show in Appendix C.2 can lead to poor predictivity of this condition.

Yaida (2018) derived a very general stationary condition that does not depend on any assumption about the underlying function $F$ and applies under general noise conditions regardless of the size of $\alpha$. Like Pflug (1983), Yaida (2018) used this condition to determine when to decrease $\alpha$, and showed good performance compared to hand-tuned SGM for two deep learning tasks with small models. However, Yaida's method does not account for the variance of the terms involved in the test, which we show can cause large variance in the learning rate schedules in some cases. This variance can in turn cause poor empirical performance.

In this work, we show how to more rigorously perform statistical hypothesis testing on samples collected from the dynamics of SGM. We combine this statistical testing with Yaida's stationary condition to develop an adaptive "constant-and-cut" optimizer (SASA) that we show is more robust than present methods. Finally, we conduct large-scale experiments on a variety of deep learning tasks to demonstrate that SASA is competitive with the best hand-tuned and validation-tuned methods without requiring additional tuning.

## 2 Stationary Conditions

To design a statistical test that fires when SGM reaches a stationary distribution, we first need to derive a condition that holds at stationarity and consists of terms that we can estimate during training. To do so, we analyze the long-run behavior of SGM with constant learning rate and momentum parameter:

$$
\begin{aligned}
d^{k+1} &= (1 - \beta)g^k + \beta d^k, \\
x^{k+1} &= x^k - \alpha d^{k+1},
\end{aligned}
\tag{2}
$$

where $\alpha > 0$ and $0 \le \beta < 1$. This process starts with $d^0 = 0$ and arbitrary $x^0$. Since $\alpha$ and $\beta$ are constant, the sequence $\{x^k\}$ does not converge to a local minimum, but the distribution of $\{x^k\}$ may converge to a stationary distribution. Letting $\mathcal{F}_k$ denote the $\sigma$-algebra defined by the history of the process (2) up to time $k$, i.e., $\mathcal{F}_k = \sigma(d^0, \ldots, d^k; x^0, \ldots, x^k)$, we denote by $\mathbf{E}_k[\cdot] := \mathbf{E}[\cdot|\mathcal{F}_k]$ the expectation conditioned on that history. Assuming that $g^k$ is Markovian and unbiased, i.e.,

$$
\mathbf{P}[g^k|\mathcal{F}_k] = \mathbf{P}[g^k|d^k, x^k], \quad \mathbf{E}[g^k|d^k, x^k] = \nabla F(x^k),
\tag{3}
$$

then the SGM dynamics (2) form a homogeneous[2] Markov chain (Bach and Moulines, 2013; Dieuleveut et al., 2017) with continuous state $(d^k, x^k, g^k) \in \mathbf{R}^{3n}$. These assumptions are always satisfied when $g^k = \nabla_x f(x^k, \xi^k)$ for an i.i.d. sample $\xi^k$. We further assume that the SGM process converges to a stationary distribution, denoted as $\pi(d, x, g)$[3]. With this notation, we need a relationship $\mathbf{E}_\pi[X] = \mathbf{E}_\pi[Y]$ for certain functions $X$ and $Y$ of $(x^k, d^k, g^k)$ that we can compute during training. Then, if we assume the Markov chain is ergodic, we have that:

$$
\bar{z}_N = \frac{1}{N} \sum_{i=0}^{N-1} z_i = \frac{1}{N} \sum_{i=0}^{N-1} \left( X(x^i, d^i, g^i) - Y(x^i, d^i, g^i) \right) \to 0.
\tag{4}
$$

Then we can check the magnitude of the time-average $\bar{z}_N$ to see how close the dynamics are to reaching their stationary distribution. Next, we consider two different stationary conditions.

### 2.1 Pflug's condition

Assuming $F(x) = (1/2)x^T A x$, where $A$ is positive definite with maximum eigenvalue $L$, and that the stochastic gradient $g^k$ satisfies $g^k = \nabla F(x^k) + r^k$, with $\mathbf{E}[r^k] = 0$ and $r^k$ independent of $x^k$, Pflug

([1983](#)) derived a stationary condition for the SGD dynamics. His condition can be extended to the SGM dynamics. For appropriate $\alpha$ and $\beta$, the generalized Pflug stationary condition says

$$\mathbf{E}_\pi\left[\langle g, d\rangle\right] = -\frac{\alpha(1-\beta)}{2(1+\beta)}\mathbf{tr}(A\Sigma_r) + O(\alpha^2),\tag{5}$$

where $\Sigma_r$ is the covariance of the noise $r$. One can estimate the left-hand-side during training by computing the inner product $\langle g^k, d^k\rangle$ in each iteration. Pflug ([1983](#)) also designed a clever estimator for the right-hand-side, so it is possible to compute estimators for both sides of (5).

The Taylor expansion in $\alpha$ used to derive (5) means that the relationship may only be accurate for small $\alpha$, but $\alpha$ is typically large in the first phase of training. This, together with the other assumptions required for Pflug's condition, are too strong to make the condition (5) useful in practice.

## 2.2 Yaida's condition

Yaida ([2018](#)) showed that as long as the stationary distribution $\pi$ exists, the following relationship holds *exactly:*

$$\mathbf{E}_\pi[\langle x, \nabla F(x)\rangle] = \frac{\alpha}{2}\frac{1+\beta}{1-\beta}\mathbf{E}_\pi[\langle d, d\rangle]$$

In particular, this holds for general functions $F$ and arbitrary values of $\alpha$. Because the stochastic gradients $g^k$ are unbiased, one can further show that:

$$\mathbf{E}_\pi[\langle x, g\rangle] = \frac{\alpha}{2}\frac{1+\beta}{1-\beta}\mathbf{E}_\pi[\langle d, d\rangle].\tag{6}$$

In the quadratic, i.i.d. noise setting of Section 2.1, the left-hand-side of (6) is simply $\mathbf{E}_\pi[x^T A x]$, twice the average loss value at stationarity. So this condition can be considered as a generalization of "testing for when the loss is stationary." We can estimate both sides of (6) by computing $\langle x^k, g^k\rangle$ and $\langle d^k, d^k\rangle = ||d^k||^2$ at each iteration and updating the running mean $\bar{z}_N$ with their difference. That is, we let

$$z_k = \langle x^k, g^k\rangle - \frac{\alpha}{2}\frac{1+\beta}{1-\beta}\langle d^k, d^k\rangle \qquad \bar{z}_N = \frac{1}{N}\sum_{k=B}^{N+B-1} z_k.\tag{7}$$

Here $B$ is the number of samples discarded as part of a "burn-in" phase to reduce bias that might be caused by starting far away from the stationary distribution; we typically take $B = N/2$, so that we use the most recent $N/2$ samples.

Yaida's condition has two key advantages over Pflug's: it holds with no approximation for arbitrary functions $F$ and any learning rate $\alpha$, and both sides can be estimated with negligible cost. In Appendix C.2, we show in Figure 14 that even on a strongly convex function, the error term in (5) is large, whereas $\bar{z}_N$ in (7) quickly converges to zero. Given these advantages, in the next section, we focus on how to test (6), i.e., that $\bar{z}_N$ defined in (7) is approximately zero.

# 3 Testing for Stationarity

By the Markov chain law of large numbers, we know that $\bar{z}_N \to 0$ as $N$ grows, but there are multiple ways to determine whether $\bar{z}_N$ is "close enough" to zero that we should drop the learning rate.

**Deterministic test.** If in addition to $\bar{z}_N$ in (7), we keep track of

$$\bar{v}_N = \frac{1}{N}\sum_{i=B}^{N+B-1}\frac{\alpha}{2}\frac{1+\beta}{1-\beta}\langle d^i, d^i\rangle,\tag{8}$$

A natural idea is to test

$$|\bar{z}_N| < \delta\bar{v}_N \quad\text{or equivalently}\quad |\bar{z}_N/\bar{v}_N| < \delta\tag{9}$$

to detect stationarity, where $\delta > 0$ is an error tolerance. The $\bar{v}_N$ term is introduced to make the error term $\delta$ relative to the scale of $\bar{z}$ and $\bar{v}$ ($\bar{v}_N$ is always nonnegative). If $\bar{z}_N$ satisfies (9), then the dynamics (2) are "close" to stationarity. This is precisely the method used by Yaida ([2018](#)).

However, because $\bar{z}_N$ is a random variable, there is some potential for error in this procedure due to its variance, which is unaccounted for by (9). Especially when we aim to make a critical decision based on the outcome of this test (i.e., dropping the learning rate), it seems important to more directly account for this variance. To do so, we can appeal to statistical hypothesis testing.

**I.i.d. $t$-test.** The simplest approach to accounting for the variance in $\bar{z}_N$ is to assume each sample $z_i$ is drawn i.i.d. from the same distribution. Then by the central limit theorem, we have that $\sqrt{N}\bar{z}_N \to \mathcal{N}(0, \sigma_z^2)$, and moreover $\hat{\sigma}_N^2 = \frac{1}{N-1}\sum_{i=1}^N (z_i - \bar{z}_N)^2 \approx \sigma_z^2$ for large $N$. So we can estimate the variance of $\bar{z}_N$'s sampling distribution using the sample variance of the $z_i$'s. Using this variance estimate, we can form the $(1-\gamma)$ confidence interval

$$\bar{z}_N \pm t^*_{1-\gamma/2} \frac{\hat{\sigma}_N}{\sqrt{N}},$$

where $t^*_{1-\gamma/2}$ is the $(1-\gamma/2)$ quantile of the Student's $t$-distribution with $N-1$ degrees of freedom. Then we can check whether

$$\left[ \bar{z}_N - t^*_{1-\gamma/2} \frac{\hat{\sigma}_N}{\sqrt{N}}, \ \bar{z}_N + t^*_{1-\gamma/2} \frac{\hat{\sigma}_N}{\sqrt{N}} \right] \in \left( -\delta\bar{v}_N, \ \delta\bar{v}_N \right). \tag{10}$$

If so, we can be confident that $\bar{z}_N$ is close to zero. The method of Pflug (1983, Algorithm 4.2) is also a kind of i.i.d. test that tries to account for the variance of $\bar{z}_N$, but in a more heuristic way than (10). The procedure (10) can be thought of as a relative *equivalence test* in statistical hypothesis testing (e.g. Streiner, 2003). When $\hat{\sigma}_N = 0$ (no variance) or $\gamma = 1$ ($t^*_{1-\gamma/2} = 0$, no confidence), this recovers (9).

Unfortunately, in our case, samples $z_i$ evaluated at nearby points are highly correlated (due to the underlying Markov dynamics), which makes this procedure inappropriate. To deal with correlated samples, we appeal to a stronger Markov chain result than the Markov chain law of large numbers (4).

**Markov chain $t$-test**   Under suitable conditions, Markov chains admit the following analogue of the central limit theorem:

**Theorem 1** (Markov Chain CLT (informal); (Jones et al., 2006)). *Let $X = \{X_0, X_1, \ldots\}$ be a Harris ergodic Markov chain with state space $\mathcal{X}$, and with stationary distribution $\pi$, that satisfies any one of a number of additional ergodicity criteria (see Jones et al. (2006), page 6). For suitable functions $z : \mathcal{X} \to \mathbb{R}$, we have that:*

$$\sqrt{N}\left( \bar{z}_N - \mathbf{E}_\pi z \right) \to \mathcal{N}(0, \sigma_z^2),$$

*where $\bar{z}_N = \frac{1}{N}\sum_{i=0}^{N-1} z(X_i)$ is the running mean over time of $z(X_i)$, and $\sigma_z^2 \neq var_\pi z$ in general due to correlations in the Markov chain.*

This shows that in the presence of correlation, the sample variance is not the correct estimator for the variance of $\bar{z}_N$'s sampling distribution. In light of Theorem 1, if we are given a consistent estimator $\hat{\sigma}_N^2 \to \sigma_z^2$, we can properly perform the test (10). All that remains is to construct such an estimator.

**Batch mean variance estimator.**   Methods for estimating the asymptotic variance of the history average estimator, e.g., $\bar{z}_N$ in (7), on a Markov chain are well-studied in the MCMC (Markov chain Monte Carlo) literature. They can be used to set a stopping time for an MCMC simulation and to determine the simulation's random error (Jones et al., 2006). We present one of the simplest estimators for $\sigma_z^2$, the *batch means* estimator.

Given $N$ samples $\{z_i\}$, divide them into $b$ batches each of size $m$, and compute the batch means: $\bar{z}^j = \frac{1}{m}\sum_{i=jm}^{(j+1)m-1} z_i$ for each batch $j$. Then let

$$\hat{\sigma}_N^2 = \frac{m}{b-1} \sum_{j=0}^{b-1} (\bar{z}^j - \bar{z}_N)^2. \tag{11}$$

Here $\hat{\sigma}_N^2$ is simply the variance of the batch means around the full mean $\bar{z}_N$. When used in the test (10), it has $b-1$ degrees of freedom. Intuitively, when $b$ and $m$ are both large enough, these batch means are roughly independent because of the mixing of the Markov chain, so their unbiased sample variance gives a good estimator of $\sigma_z^2$. Jones et al. (2006) survey the formal conditions under which $\hat{\sigma}_N^2$ is a strongly consistent estimator of $\sigma_z^2$, and suggest taking $b = m = \sqrt{N}$ (the theoretically correct sizes of $b$ and $m$ depend on the mixing of the Markov chain). Flegal and Jones (2010) prove strong consistency for a related method called *overlapping batch means* (OLBM) that has better asymptotic variance. The OLBM estimator is similar to (11), but uses $n-b+1$ overlapping batches of size $b$ and has $n-b$ degrees of freedom.

| **Algorithm 2:** SASA | **Algorithm 3:** Test |
|---|---|
| **Input:** $\{x^0, \alpha_0, M, \beta, \delta, \gamma, \zeta\}$ | **Input:** $\{zQ, vQ, \delta, \gamma\}$ |
|  | **Output:** boolean (whether to drop) |

**Algorithm 2:** SASA

**Input:** $\{x^0, \alpha_0, M, \beta, \delta, \gamma, \zeta\}$

1   zQ = HalfQueue()
2   vQ = HalfQueue()
3   **for** $j \in \{0, 1, 2, \ldots\}$ **do**
4     **for** $k \in \{jM, \ldots, (j+1)M - 1\}$ **do**
5       Sample $\xi^k$ and compute $g^k = \nabla_x f(x^k, \xi^k)$
6       $d^{k+1} = (1 - \beta)g^k + \beta d^k$
7       $x^{k+1} = x^k - \alpha d^{k+1}$
8       zQ.push($\langle x^k, g^k \rangle - \frac{\alpha}{2}\frac{1+\beta}{1-\beta}\|d^{k+1}\|^2$)
9       vQ.push($\frac{\alpha}{2}\frac{1+\beta}{1-\beta}\|d^{k+1}\|^2$)
10     **end**
11     **if** $test(zQ, vQ, \delta, \gamma)$ **then**
12       $\alpha \leftarrow \zeta\alpha$
13       zQ.reset()
14       vQ.reset()
15     **end**
16   **end**

**Algorithm 3:** Test

**Input:** $\{zQ, vQ, \delta, \gamma\}$
**Output:** boolean (whether to drop)

1   $\bar{z}_N = \frac{1}{zQ.N} \sum_i zQ[i]$
2   $\bar{v}_N = \frac{1}{vQ.N} \sum_i vQ[i]$
3   $m = b = \sqrt{zQ.N}$
4   **for** $i \in \{0, \ldots, b-1\}$ **do**
5     $\bar{z}^i = \frac{1}{m} \sum_{t=im}^{(i+1)m-1} zQ[t]$
6   **end**
7   $\hat{\sigma}_N^2 = \frac{m}{b-1} \sum_{i=0}^{b-1} (\bar{z}^i - \bar{z}_N)^2.$
8   $L = \bar{z}_N - t^*_{1-\gamma/2} \frac{\hat{\sigma}_N}{\sqrt{zQ.N}}$
9   $U = \bar{z}_N + t^*_{1-\gamma/2} \frac{\hat{\sigma}_N}{\sqrt{zQ.N}}$
10   **return** $[L, U] \in (-\delta\bar{v}_N, \delta\bar{v}_N)$

## 3.1 Statistical adaptive stochastic approximation (SASA)

Finally, we turn the above analysis into an adaptive algorithm for detecting stationarity of SGM and decreasing $\alpha$, and discuss implementation details. Algorithm 2 describes our full SASA algorithm.

To diminish the effect of "initialization bias" due to starting outside of the stationary distribution, we only keep track of the latter half of samples $z_i$ and $v_i$. That is, if $N$ total iterations of SGM have been run, the "HalfQueues" $zQ$ and $vQ$ contain the most recent $N/2$ values of $z_i$ and $v_i$—these queues "pop" every other time they "push." If we decrease the learning rate, we empty the queues; otherwise, we keep collecting more samples. To compute the batch mean estimator, we need $O(N)$ space, but in deep learning the total number of training iterations (the worst case size of these queues) is usually small compared to the number of parameters of the model. Collection of the samples $z_i$ and $v_i$ only requires two more inner products per iteration than SGM.

The "test" algorithm follows the Markov chain $t$-test procedure discussed above. Lines 1-2 compute the running means $\bar{z}_N$ and $\bar{v}_N$; lines 3-7 compute the variance estimator $\hat{\sigma}_N^2$ according to (11), and lines 8-10 determine whether the corresponding confidence interval for $\bar{z}_N$ is within the acceptable interval $(-\delta\bar{v}_N, \delta\bar{v}_N)$. Like the sample collection, the test procedure is computationally efficient: the batch mean and overlapping batch mean estimators can both be computed with a 1D convolution.

For all experiments, we use default values $\delta = 0.02$ and $\gamma = 0.2$. In equivalence testing, $\gamma$ is typically taken larger than usual to increase the power of the test (Streiner, 2003). We discuss the apparent multiple testing problem of this sequential testing procedure in Appendix D.

## 4 Experiments

To evaluate the performance of SASA, we run Algorithm 2 on several models from deep learning. We compare SASA to *tuned* versions of Adam and SGM. Many adaptive optimizers do not compare to SGM with hand-tuned step size scheduling, (e.g., Schaul et al., 2013; Zhang and Mitliagkas, 2017; Baydin et al., 2018), and instead compare to SGM with a fixed $\alpha$ or to SGM with tuned polynomial decay. As detailed in Section 1, tuned constant-and-cut schedules are typically a stronger baseline.

Throughout this section, we *do not tune the SASA parameters $\delta, \gamma, M$*, instead using the default settings of $\delta = 0.02$ and $\gamma = 0.2$, and setting $M =$ one epoch (we test the statistics once per epoch). In each experiment, we use the same $\alpha_0$ and $\zeta$ as for the best SGM baseline. We stress that SASA is not fully automatic: it requires choices of $\alpha_0$ and $\zeta$, but we show in Appendix A that SASA achieves good performance for different values of $\zeta$. We use weight decay in every experiment—without weight decay, there are simple examples where the process (2) does not converge to a stationary distribution,

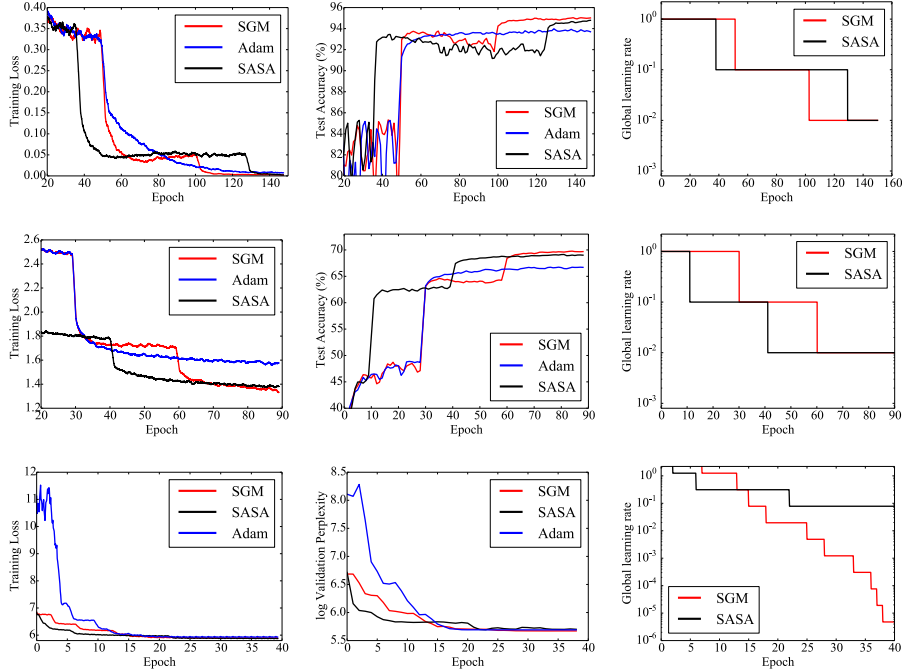

Figure 2: Training loss, test accuracy, and learning rate schedule for SASA, SGM, and Adam on different datasets. Top: ResNet18 on CIFAR-10. Middle: ResNet18 on ImageNet. Bottom: RNN model on WikiText-2. In all cases, starting with the same $\alpha_0$, SASA achieves similar performance to the best hand-tuned or validation-tuned SGM result. Across three independent runs, the variance of each optimizer's best test accuracy was never larger than 1%, and the relative orderings between optimizers held for every run. Figure 5 studies the variance of SASA in a semi-synthetic setting.

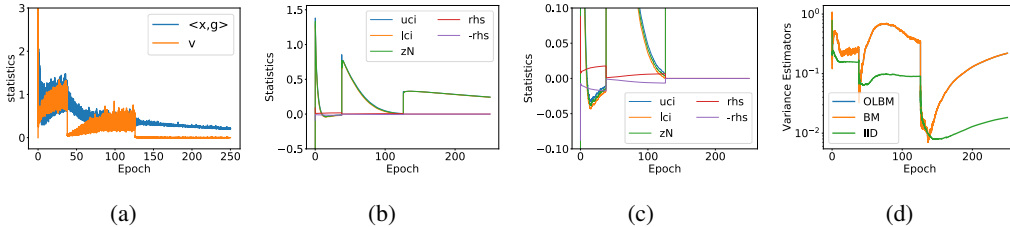

(a)  (b)  (c)  (d)

Figure 3: Evolution of the different statistics for SASA over the course of training ResNet18 on CIFAR-10 using the default parameters $\delta = 0.02, \gamma = 0.2, \zeta = 0.1$. Panel (a) shows the raw data for both sides of condition (6). That is, it shows the values of $\langle x^k, g^k \rangle$ and $\frac{\alpha}{2}\frac{1+\beta}{1-\beta}\langle d^k, d^k \rangle$ at each iteration. Panel (1) shows $\bar{z}_N$ with its lower and upper confidence interval [$lci, uci$] and the "right hand side" (rhs) $(-\delta\bar{v}_N, \delta\bar{v}_N)$ (see Eqn. (10)). Panel (c) shows a zoomed-in version of (b) to show the drop points in more detail. Panel (d) depicts the different variance estimators (i.i.d., batch means, overlapping batch means) over the course of training. The i.i.d. variance (green) is a poor estimate of $\sigma_z^2$.

such as with logistic regression on separable data. While weight decay does not guarantee convergence to a stationary distribution, it at least rules out this simple case. Finally, we conduct an experiment on CIFAR-10 that shows directly accounting for the variance of the test statistic, as in (10), improves the robustness of this procedure compared to (9).

For hand-tuned SGM (SGM-hand), we searched over "constant-and-cut" schemes for each experiment by tuning $\alpha_0$, the drop frequency, and the drop amount $\zeta$ with grid search. In all experiments, SASA and SGM use a constant $\beta = 0.9$. For Adam, we tuned the initial global learning rate as in Wilson et al. (2017) and used $\beta_1 = 0.9, \beta_2 = 0.999$. We also allowed Adam to have access to a "warmup" phase to prevent it from decreasing the learning rate too quickly. To "warm up" Adam, we initialize it with the

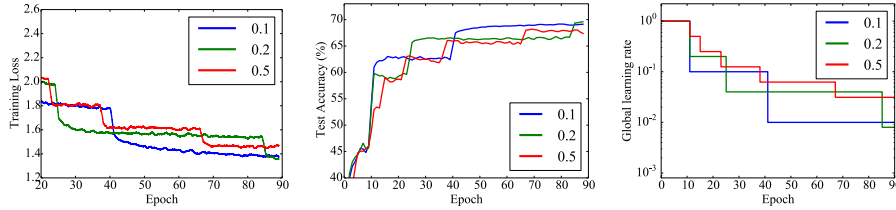

Figure 4: Smoothed training loss, test accuracy and learning rate schedule for ResNet18 trained on ImageNet using SASA with different values of $\zeta$. SASA automatically adapts the drop frequency.

parameters obtained after running SGM with constant $\alpha_0$ for a tuned number of iterations. While the warmup phase improves Adam's performance, it still does not match SASA or SGM on the tasks we tried. Appendix A contains a full list of the hyperparameters used in each experiment, additional results for object detection, sensitivity analysis for $\delta$ and $\gamma$, and plots of the different estimators for the variance $\sigma_z^2$.

**CIFAR-10.** We trained an 18-layer ResNet model[4] He et al. (He et al., 2016) on CIFAR-10 (Krizhevsky and Hinton, 2009) with random cropping and random horizontal flipping for data augmentation and weight decay 0.0005. Row 1 of Figure 2 compares the best performance of each method. Here SGM-hand uses $\alpha_0 = 1.0$ and $\beta = 0.9$ and drops $\alpha$ by a factor of 10 ($\zeta = 0.1$) every 50 epochs. SASA uses $\gamma = 0.2$ and $\delta = 0.02$, as always. Adam has a tuned global learning rate $\alpha_0 = 0.0001$ and a tuned "warmup" phase of 50 epochs, but is unable to match SASA and tuned SGM.

**Evolution of statistics.** Figure 3 shows the evolution of SASA's different statistics over the course of training the ResNet18 model on CIFAR-10 using the default parameter settings $\delta = 0.02, \gamma = 0.2, \zeta = 0.1$. In each phase, the running average of the difference between the statistics, $\bar{z}_N$, decays toward zero. The learning rate $\alpha$ drops once $\bar{z}_N$ and its confidence interval are contained in $(-\delta \bar{v}_N, \delta \bar{v}_N)$; see Eqn (10). After the drop, the statistics increase in value and enter another phase of convergence. The batch means variance estimator (BM) and overlapping batch means variance estimator (OLBM) give very similar estimates of the variance, while the i.i.d. variance estimator, as expected, gives quite different values.

**ImageNet.** Unlike CIFAR-10, reaching a good performance level on ImageNet (Deng et al., 2009) seems to require more gradual annealing. Even when tuned and allowed to have a long warmup phase, Adam failed to match the generalization performance of SGM. On the other hand, SASA was able to match the performance of hand-tuned SGM using the default values of its parameters. We again used an 18-layer ResNet model with random cropping, random flipping, normalization, and weight decay 0.0001. Row 2 of Figure 2 shows the performance of the different optimizers.

**RNN.** We also evaluate SASA on a language modeling task using an RNN. In particular, we train the PyTorch word-level language model example (2019) on the Wikitext-2 dataset (Merity et al., 2016). We used 600-dimensional embeddings, 600 hidden units, tied weights, and dropout 0.65, and gradient clipping with threshold 2.0 (note that this model is not state-of-the-art for Wikitext-2). We compare against SGM and Adam with (global) learning rate tuned using a validation set. These baselines drop the learning rate $\alpha$ by a factor of 4 when the validation loss stops improving. Row 3 of Figure 2 shows that *without using the validation set*, SASA is competitive with these baselines.

**Adaptation to the drop factor.** At first glance, the choice of the drop factor $\zeta$ seems critical. However, Figure 4 shows that SASA automatically adapts to different values of $\zeta$. When $\zeta$ is larger, so $\alpha$ decreases slower, the dynamics converge more quickly to the stationary distribution, so the overall rate of decrease stays roughly constant across different values of $\zeta$. Aside from the different choices of $\zeta$, all other hyperparameters were the same as in the ImageNet experiment of Figure 2.

**Variance.** Figure 5 shows the variance in learning rate schedule and training loss for the two tests in (9) (top row) and (10) (bottom row) with a fixed testing frequency $M = 400$ iterations, across five independent runs. The model is ResNet18 trained on CIFAR-10 using the same procedure as

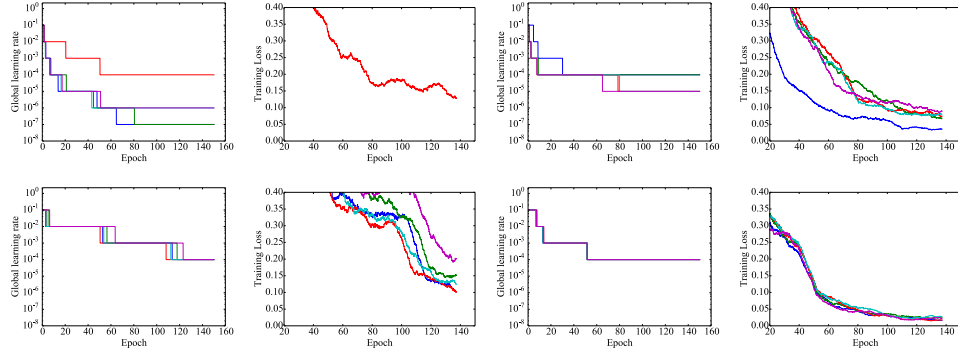

Figure 5: Variance in learning rate schedule and training loss for the two tests (9) (top row) and (10) (bottom row) with *fixed* testing frequency $M$, across five independent runs. The left two columns use batch size four, and the right two use batch size eight. With the same testing frequency and the same value of $\delta$ (0.02), the test (9) is much more sensitive to the level of noise. In row 1, column 2, only one of the five runs (plotted in red) achieves a low training loss because of the high variance in schedule (row 1, column 1).

in the previous CIFAR experiment, but with different batch sizes. The left two columns use batch size four, and the right two use batch size eight. With the same testing frequency and the same value of $\delta = 0.02$, the test (9) is much more sensitive to the level of noise in these small-batch examples. When the batch size is four, only one of the training runs using the test (9) achieves training loss on the same scale as the others. Appendix B contains additional discussion comparing these two tests.

## 5 Conclusion

We provide a theoretically grounded statistical procedure for automatically determining when to decrease the learning rate $\alpha$ in constant-and-cut methods. On the tasks we tried, SASA was competitive with the best hand-tuned schedules for SGM, and it came close to the performance of SGM and Adam when they were tuned using a validation set. The statistical testing procedure controls the variance of the method and makes it more robust than other more heuristic tests. Our experiments across several different tasks and datasets did not require any adjustment to the parameters $\gamma$, $\delta$, or $M$.

We believe these practical results indicate that automatic "constant-and-cut" algorithms are a promising direction for future research in adaptive optimization. We used a simple statistical test to check Yaida's stationary condition (6). However, there may be better tests that more properly control the false discovery rate (Blanchard and Roquain, 2009; Lindquist and Mejia, 2015), or more sophisticated conditions that also account for non-stationary dynamics like overfitting or limit cycles (Yaida, 2018). Such techniques could make the SASA approach more broadly useful.

## Footnotes

[1]For fixed values of $\alpha$ and $\beta$, this "normalized" update formula is equivalent to the more common updates $d^{k+1} = g^k + \beta d^k$, $x^{k+1} = x^k - \alpha' d^{k+1}$ with the reparametrization $\alpha = \alpha'/(1 - \beta)$.

[2]"Homogeneous" means that the transition kernel is time independent.

[3]As stated in Section 1, this need not be true in general, but seems to often be the case in practice.

[4]In our experiments on CIFAR-10, we used the slightly modified ResNet model of `https://github.com/kuangliu/pytorch-cifar`, which we found to give a small performance gain over the model of He et al. (2016) for all optimizers we tested. The first convolutional layer in this model uses filter size 3 with stride 1 and padding 1, rather than 7, 2, and 3, respectively.

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
