[Supplementary Material]



Figure 5: Evolution of the different statistics for SASA over the course of training ResNet18 on CIFAR-10 using the default parameters $\delta = 0.02, \gamma = 0.2, \zeta = 0.1$. Panel (a) shows the raw data for both sides of condition (6). That is, it shows the values of $\langle x^k, g^k \rangle$ and $\frac{\alpha}{2} \frac{1+\beta}{1-\beta} \langle d^k, d^k \rangle$ at each iteration. Panel (1) shows $\bar{z}_N$ with its lower and upper confidence interval $[lci, uci]$ and the "right hand side" (rhs) $(-\delta \bar{v}_N, \delta \bar{v}_N)$ (see Eqn. (10)). Panel (c) shows a zoomed-in version of (b) to show the drop points in more detail. Panel (d) depicts the different variance estimators (i.i.d., batch means, overlapping batch means) over the course of training. The i.i.d. variance (green) is a poor estimate of $\sigma_z^2$.

Figure 6: Training loss, test accuracy, and learning rate schedule for SASA using different values of $\gamma$, $\delta$ and $\zeta$ around the defaults 0.2, 0.02 and 0.1. The model is ResNet18 trained on CIFAR-10, with the procedure the same as in Section 4. Top row: performance for fixed $\gamma = 0.2, \zeta = 0.1$, and $\delta \in \{0.005, 0.01, 0.02, 0.04\}$. Middle row: performance for fixed $\delta = 0.02, \zeta = 0.1$, and $\gamma \in \{0.05, 0.1, 0.2\}$. Bottom row: performance for fixed $\gamma = 0.2, \delta = 0.02$, and $\zeta \in \{0.5, 0.2, 0.1\}$. Qualitatively, increasing $\delta$ and increasing $\gamma$ both cause the algorithm to drop sooner. The value of $\zeta$ does not influence the final performance, as long as the learning rate finally decays to the same level.

## A  Details and Additional Experiments

In this section, we provide more experimental details and discussion, and examine the sensitivity of SASA's performance with respect to its parameters.

## A.1 CIFAR-10 experiments in Section 4

For the CIFAR-10 experiment, we set SGM to use $\alpha_0 = 1.0$ and $\zeta = 0.1$, and drop every 50 epochs. We used the same values of $\alpha_0$ and $\zeta$ for SASA. For Adam, we used a warmup phase of 50 epochs with $\alpha = 1.0$, and then set $\alpha_0 = 0.0001$, the optimal value in our grid search of $\{0.00001, 0.0001, 0.01, 0.1\}$. The weight decay parameter for all models was set to 0.0005. All methods used batch size 128.

**Evolution of statistics.** Figure 5 shows the evolution of SASA's different statistics over the course of training the ResNet18 model on CIFAR-10 using the default parameter settings $\delta = 0.02, \gamma = 0.2, \zeta = 0.1$. In each phase, the running average of the difference between the statistics, $\bar{z}_N$, decays toward zero. The learning rate $\alpha$ drops once $\bar{z}_N$ and its confidence interval are contained in $(-\delta\bar{v}_N, \delta\bar{v}_N)$; see Eqn (10). After the drop, the statistics increase in value and enter another phase of convergence. The batch means variance estimator (BM) and overlapping batch means variance estimator (OLBM) give very similar estimates of the variance, while the i.i.d. variance estimator, as expected, gives quite different values.

**Sensitivity analysis.** We perturb the relative equivalence threshold $\delta$, the confidence level $\gamma$ and the decay rate $\zeta$ around their default values $(0.2, 0.02, 0.1)$ and repeat the CIFAR-10 experiment from the previous section, using the same values for the other hyperparameters. In Figure 6, the top row shows the performance for fixed $(\gamma, \zeta) = (0.2, 0.1)$ and changing $\delta$. The middle row shows the performance for fixed $(\delta, \zeta) = (0.02, 0.1)$ and changing $\gamma$. The bottom row shows the performance for fixed $(\delta, \gamma) = (0.02, 0.2)$ and changing $\zeta$. Increases in both $\delta$ and $\gamma$ tend to cause the algorithm to drop sooner; this behavior is intuitive from the testing procedure (10). For values of the parameters close to the defaults, SASA still obtains good performance. The value of $\zeta$ does not influence the final performance, as long as the learning rate finally decays to the same level.

## A.2 ImageNet experiments in Section 4

For the ImageNet experiment, we again used $\alpha_0 = 1.0$ and $\zeta = 0.1$ for SGM and SASA, and dropped the SGM learning rate every 30 epochs. We let Adam have a warmup phase of 30 epochs, initializing it with the parameters obtained from running SGM with $\alpha = 1.0$. After this phase, we used $\alpha_0 = 0.0001$, the optimal value from a grid $\{0.00001, 0.0001, 0.001, 0.01\}$. The weight decay for all models was set to 0.0001. All methods used batch size 256.

**Evolution of statistics.** Figure 7 shows the evolution of SASA's different statistics over the course of training the ResNet18 model on CIFAR-10, under the default parameter setting $\delta = 0.02, \gamma = 0.2, \zeta = 0.1$. In each phase, $z$ and $v$ get close two easy other as predicted by (6). Together with its confidence interval, the statistics $\bar{z}_N$ decay toward zero. The learning rate is dropped as long as the confidence interval is contained in $(-\delta\bar{v}_N, \delta\bar{v}_N)$; see Eqn (10). The batch mean variance estimator (bm) and overlapping batch mean variance estimator (olbm) give very close variance estimates, while the i.i.d. variance estimator is clearly much different from the batch mean and overlapping batch mean estimators.

**Sensitivity analysis.** We perturb the relative equivalence threshold $\delta$, the confidence level $\gamma$ and the decay rate $\zeta$ around their default values $(0.2, 0.02, 0.1)$ and repeat the CIFAR-10 experiment from the previous section, using the same values for the other hyperparameters. In Figure 8, the top row shows the performance for fixed $(\gamma, \zeta) = (0.2, 0.1)$ and changing $\delta$. The middle row shows the performance for fixed $(\delta, \zeta) = (0.02, 0.1)$ and changing $\gamma$. The bottom row shows the performance for fixed $(\delta, \gamma) = (0.02, 0.2)$ and changing $\zeta$. Increases in both $\delta$ and $\gamma$ tend to cause the algorithm to drop sooner; this behavior is intuitive from the testing procedure (10). For values of the parameters close to the defaults, SASA still obtains good performance. The value of $\zeta$ does not influence the final performance, as long as the learning rate finally decays to the same level.

## A.3 RNN experiments in Section 4

For the RNN experiment, we trained the PyTorch word-level language model example (2019) with 600 hidden units, 600-dimensional embeddings, dropout 0.65, and tied weights. All optimizers also used gradient clipping with 2.0 as the threshold and weight decay 0.0005. We set $\alpha_0$ and $\zeta$ for SGM and SASA to be 2.0 and 0.25, respectively. Because Adam was tuned in this example using the

Figure 7: Evolution of the different statistics for SASA over the course of training ResNet18 on ImageNet using the default parameters $\delta = 0.02, \gamma = 0.2, \zeta = 0.1$. Panel (a) shows the raw data for both sides of condition (6). That is, it shows the values of $\langle x^k, g^k \rangle$ and $\frac{\alpha}{2} \frac{1+\beta}{1-\beta} \langle d^k, d^k \rangle$ at each iteration. Panel (b) shows $\bar{z}_N$ with its lower and upper confidence interval $[lci, uci]$ and the "right hand side" (rhs) $(-\delta \bar{v}_N, \delta \bar{v}_N)$ (see Eqn. (10)). Panel (c) shows a zoomed-in version of (b) to show the drop points in more detail. Panel (d) depicts the different variance estimators (i.i.d., batch means, overlapping batch means) over the course of training. The i.i.d. variance (green) is a poor estimate of $\sigma_z^2$.

Figure 8: Training loss, test accuracy, and learning rate schedule for SASA using different values of $\gamma$, $\delta$ and $\zeta$ around the default 0.2, 0.02 and 0.1. The model is ResNet18 trained on ImageNet, as in 4. Top row: performance for fixed $\gamma = 0.2, \zeta = 0.1$, and $\delta \in \{0.005, 0.01, 0.02\}$. Middle row: performance for fixed $\delta = 0.02, \zeta = 0.1$, and $\gamma \in \{0.05, 0.1, 0.2\}$. Bottom row: performance for fixed $\gamma = 0.2, \delta = 0.02$, and $\zeta \in \{0.5, 0.2, 0.1\}$. Qualitatively, increasing $\delta$ and increasing $\gamma$ both cause the algorithm to drop sooner. The value of $\zeta$ does not influence the final performance, as long as the learning rate finally decays to the same level.

validation set, we also used $\zeta = 0.25$ for Adam. The optimal $\alpha_0$ for Adam was 0.5, chosen from the grid $\{0.1, 0.5, 1.0, 2.0, 3.0\}$.

## A.4  Additional experiment: training logistic regression on the MNIST dataset

We train a logistic regression model on the MNIST dataset with weight decay 0.0005.

Figure 9: Top: training loss, test accuracy, and learning rate schedule for SASA and Adam for logistic regression on MNIST. Bottom: Evolution of the different statistics for SASA, as in Figures 5 and 7. SASA uses its default parameters $(\delta, \gamma, \zeta) = (0.02, 0.2, 0.1)$. Adam uses its default $(\beta_1, \beta_2) = (0.9, 0.999)$ but its initial learning rate $\alpha_0 = 0.00033$ is obtained from a grid search.

**Default value performance.** Figure 9 shows SASA's performance with default parameters. For this convex optimization problem, SASA and Adam achieve similar performance. SASA uses its default parameters $(\delta, \gamma, \zeta) = (0.02, 0.2, 0.1)$ and initial $\alpha_0 = 1.0$. Adam uses its default $(\beta_1, \beta_2) = (0.9, 0.999)$ and its initial learning rate $lr = 0.00033$ is obtained from a grid search over $\{0.01, 0.0033, 0.001, 0.00033, 0.0001\}$.

**Sensitivity analysis.** As with the experiments on CIFAR-10 and ImageNet, we perturb the relative equivalence threshold $\delta$, the confidence level $\gamma$, and the decay rate $\zeta$ around their default values $(0.2, 0.02, 0.1)$. In Figure 10, the top row shows the performance for fixed $(\gamma, \zeta) = (0.2, 0.1)$ and changing $\delta$. The middle row shows the performance for fixed $(\delta, \zeta) = (0.02, 0.1)$ and changing $\gamma$. The bottom row shows the performance for fixed $(\delta, \gamma) = (0.02, 0.2)$ and changing $\zeta$. The results are the qualitatively the same as in Figures 6 and 8.

## A.5 Additional experiment: training MaskRCNN on the COCO dataset

We train a Mask-RCNN model He et al. (2017) with a Feature Pyramid Network (FPN) Lin et al. (2017) as a backbone for both object detection and instance segmentation on the the COCO dataset Lin et al. (2014). The FPN backbone is based on the ResNet50, and the implementation is based on the MaskRCNN-benchmark repo Massa and Girshick (2018). In the recommend training setting, the model is trained for 90000 iterations with the SGM optimizer. The learning rate is scheduled to decay by 10 ($\zeta = 0.1$) at iteration 60000 and 80000. Readers can refer to `https://github.com/facebookresearch/maskrcnn-benchmark/blob/master/configs/e2e_mask_rcnn_R_50_FPN_1x.yaml` for a detailed experiment setup. This hyperparameter setting is carefully tuned to reach the reported performance: object detection mean average precision (bbox-AP) 37.8% and instance segmentation mean average precision (segm-AP) 34.2%; see `https://github.com/facebookresearch/maskrcnn-benchmark/blob/master/MODEL_ZOO.md`.

**Default value performance.** Figure 11 shows SASA's performance with default parameters. For this challenging task, SASA achieves a slightly better performance than the hand-tuned SGM optimizer *without any parameter tuning*. However, SASA with default parameters takes longer to achieve comparable performance, because SASA decides to decay the learning rate later than the hand-tuned SGM. Notice that SASA only decreases the learning rate once and already surpasses the performance of the hand-tuned SGM. We believe that if the learning rate is decreased again, the performance can be further improved. However, when the training reaches the maximum iteration 200000, the training loss is still constantly decreasing, so the dynamics have not reached a stationary distribution. This prevents SASA from decreasing its learning rate. Meanwhile, the model starts to overfit at this stage, which suggests that we should either decrease the learning rate or stop the training. As mentioned in Section 5, a combination of stationary detection (SASA) and overfitting detection is a promising direction toward a fully automated optimizer.

Figure 10: Training loss, test accuracy, and learning rate schedule for SASA using different values of $\gamma$, $\delta$ and $\zeta$ around the default 0.2, 0.02 and 0.1. The model is the logistic regression trained on MNIST. Top row: performance for fixed $\gamma = 0.2$, $\zeta = 0.1$, and $\delta \in \{0.005, 0.01, 0.02, 0.04\}$. Middle row: performance for fixed $\delta = 0.02$, $\zeta = 0.1$, and $\gamma \in \{0.05, 0.1, 0.2\}$. Bottom row: performance for fixed $\gamma = 0.2$, $\delta = 0.02$, and $\zeta \in \{0.5, 0.2, 0.1\}$. Qualitatively, increasing $\delta$ and increasing $\gamma$ both cause the algorithm to drop sooner. The value of $\zeta$ does not influence the final performance, as typically the learning rate automatically decays to the same level.

## B   Comparison with Yaida's test

The variance experiment in Figure 4 can be interpreted as showing that for a fixed testing frequency $M$, the statistical procedure (10) is more robust to changes in the noise level of the samples than the heuristic test (9). We essentially repeat this experiment in Figure 12, which shows the performance of the two testing methods (9) and (10) on a logistic regression model trained on MNIST. We used the same procedure as in all other logistic regression experiments, except with batch size one. We test the statistics every $M = 100$ iterations and plot the results of ten independent runs for each method, using a fixed $M$ as in Figure 4. While the final training and test loss for the two methods are similar (92.7% $\pm 0.16$ for SASA, 92.7% $\pm 0.2$ for (9)), the variance in the learning rate schedules for Yaida's method is dramatically higher. On strongly convex problems, this may not cause poor performance, but as shown in Figure 4, it can cause dramatic results in more general settings. This experiment gives a further indication that when using a fixed test frequency $M$, explicitly accounting for the variance in $\bar{z}_N$, as in SASA, is critical for robust performance. Finally, Figure 13 shows a complementary result on CIFAR-10: even when the batch size is large (128), the statistical approach is less sensitive to using a small testing frequency $M$. While this effect is on a much smaller scale than the others, it indicates that Yaida's heuristic (performing the test (9) once per epoch) is more sensitive than SASA to the choice of the testing frequency.

Figure 4, Figure 12, and Figure 13 indicate that the statistical test is more robust to changes in noise and testing frequency than Yaida's deterministic ratio test. However, Figure 13 indicates that this method can obtain similar (albeit less robust) performance on large deep learning datasets, and our practical results can be taken more generally as large-scale evidence that methods for detecting stationarity have good practical performance when used as adaptive optimizers. Still, our formulation recovers Yaida's when $\gamma = 1$, heuristics like "test once per epoch" are not always available—such as in an online training setting—so robustness to the test frequency $M$ is desirable, and we have

Figure 11: Top: training loss, test accuracy, and learning rate schedule for SASA and SGM for MaskRCNN training on COCO. Bottom: Evolution of the different statistics for SASA, as in Figures 5 and 7. SASA uses its default parameters $(\delta, \gamma, \zeta) = (0.02, 0.2, 0.1)$. The SGM is scheduled to decay the learning rate by 10 ($\zeta = 0.1$) twice, once at iteration 60000 and once at iteration 90000. SASA takes more iterations (double) to reach a slightly better performance without any parameter tuning.

|   (a)   |   (b)   |   (c)   |   (d)   |

Figure 12: Variance in learning rate schedule and training loss for the two tests (9) (Panels (a)-(b)) and (10) (Panels (c)-(d)) for a logistic regression model on MNIST, using batch size one and test frequency $M = 100$ iterations. Ten independent runs are shown for each method. With the same value of $\delta$, the variance in the learning rate schedule for Yaida's method (9) is much higher.

demonstrated that SASA is less sensitive to noise in several regimes, such as small batch size and high test frequency. For these reasons we believe SASA will be more robust in practice, and we hope it leads to more research on using statistical tests in optimization.

## C Generalized Pflug condition and comparison with Yaida's condition

In this section, we provide a generalization of Pflug's stationary condition to the case of SGM for quadratic functions. We also compare the two stationary conditions (Pflug's and Yaida's) and show that Yaida's stationary condition works much better for practical machine learning problems.

### C.1 Derivation of the generalized Pflug stationary condition

As in (Pflug, 1990; Mandt et al., 2017), the derivation is based on two assumptions:

1. The quadratic objective assumption:

$$F(x) = (1/2)x^T A x, \tag{12}$$

where $A$ is positive definite.

2. The i.i.d. additive noise assumption:

$$g^k = \nabla F(x^k) + \xi^k, \tag{13}$$

where $\xi^k$ is independent of $x^k$, and for all $k \geq 0$ satisfies

$$\mathbf{E}\left[\xi^k\right] = 0, \qquad \mathbf{E}\left[\xi^k(\xi^k)^T\right] = \Sigma_\xi. \tag{14}$$

Figure 13: Test accuracy and learning rate schedule when using Yaida's ratio test (9) (top row) and our statistical test (10) (bottom row) with $M = 10$, $\delta = 0.02$, and with SASA using $\gamma = 0.2$. The standard deviation of both the learning rate drops and the best test set performance is higher for Yaida's test: 3.2 epochs vs 2.9 epochs, and 0.17% vs 0.12%. The mean performance of the statistical test is also marginally higher, 94.08% vs 93.84% test accuracy.

Mandt et al. (2017) observe that this noise assumption can hold approximately when $\alpha$ is small and the dynamics of SGM are approaching stationarity around a local minimum.

For the dynamics of SGM with constant $\alpha$ and $\beta$, i.e., (2), the sequence $\{(x^k, d^k, g^k)\}$ is assumed to converge to a stationary distribution $\pi(x, d, g)$, as we defined in Section 2. We denote $x$'s covariance matrix under the stationary distribution as

$$\Sigma_x = \lim_{k \to \infty} \mathbf{E}\left[x^k (x^k)^T\right].$$ (15)

The following theorem characterizes the dependence of $\Sigma_x$ on $A$, $\alpha$ and $\beta$. It also derives an asymptotic expression of $\mathbf{E}_\pi[\langle g, d \rangle]$ in terms of $A$, $\alpha$ and $\beta$.

**Theorem 2.** *Suppose $F(x) = (1/2)x^T A x$, where $A$ is positive definite with maximum eigenvalue $L$, and $g^k$ satisfies (13) and (14). If we choose $\alpha \in (0, 1/L)$ and $\beta \in [0, 1)$ in (2), then $\Sigma_x$ defined in (15) exists. Moreover, we have*

$$A\Sigma_x + \Sigma_x A = \alpha \Sigma_\xi + O(\alpha^2)$$ (16)

*and*

$$\mathbf{E}_\pi[\langle g, d \rangle] = -\frac{\alpha(1 - \beta)}{2(1 + \beta)} \mathbf{tr}(A\Sigma_\xi) + O(\alpha^2).$$ (17)

Theorem 2 states that when $\alpha$ is small, we can approximate $\Sigma_x$ by solving the linear equation $A\Sigma_x + \Sigma_x A = \Sigma_\xi$. Moreover, the variance $\mathbf{tr}(\Sigma_x)$ decreases to zero as $\alpha \to 0$. It is well known that larger $\beta$ often leads to faster transient convergence when SGM is far away from a local minimum. According to (16), it does not affect the covariance in steady state, especially for small $\alpha$.

Equation (17) implies that for small $\alpha$, the vectors $g^k$ and $d^k$ will eventually have negative correlation. Interestingly, their correlation is less negative for larger $\beta$.

Assuming ergodicity, $\mathbf{E}_\pi[\langle g, d \rangle]$ can be evaluated by the history average

$$\mathbf{E}_\pi\left[\langle g^k, d^k \rangle\right] \approx \frac{1}{N} \sum_{i=k+1}^{k+N} \langle g^i, d^i \rangle, \tag{18}$$

where $N$ can be chosen to control the quality of estimation. If an online estimate of $\mathbf{tr}(A\Sigma_\xi)$ is also available, then we can check if the relation established in (17) holds in a statistical sense, which serves as a test of stationarity.

Since we do not assume any knowledge of $A$ or $\Sigma_\xi$, it can be hard to estimate $\mathbf{tr}(A\Sigma_\xi)$ using simple statistics. To address this challenge, Pflug (1983) constructed a novel scheme that requires three stochastic gradients at each iteration. Specifically, at each iteration $k$, we first compute two stochastic gradients $g_1^k$ and $g_2^k$ of $F$ at $x^k$, and we let $r^k = (g_1^k - g_2^k)/2$ (in a data-parallel training setting, $g_1^k$ and $g_2^k$ can be computed from separate processing units, and thus can be obtained without extra delay). Next, we let $\tilde{x}^k = x^k + \alpha r^k$ and compute another stochastic gradient $\tilde{g}^k$ of $F$ at $\tilde{x}^k$. Then, it can be shown (Pflug, 1983) that

$$\mathbf{E}[\langle r^k, \tilde{g}^k \rangle] = \frac{\alpha}{2}\mathbf{tr}(A\Sigma_\xi). \tag{19}$$

We can thus obtain an online estimate of $\mathbf{tr}(A\Sigma_\xi)$ using the running average of $\langle r^k, \tilde{g}^k \rangle$ in a similar way to (18).

As suggested by Pflug (1983), a less wasteful use of the stochastic gradients is to define $g^k = (g_1^k + g_2^k)/2$ and use it in (2). This averaging reduces the covariance of $g^{k+1}$ and $d^{k+1}$ by a factor of $1/2$, which together with (17) implies

$$\mathbf{E}_\pi\left[\langle g, d \rangle\right] \approx -\frac{\alpha(1-\beta)}{4(1+\beta)}\mathbf{tr}(A\Sigma_\xi), \tag{20}$$

where we still use $\pi$ to denote the new stationary condition. Combining (19) and (20), we conclude that for small $\alpha$,

$$\mathbf{E}_\pi\left[\langle g, d \rangle\right] \approx -\frac{1-\beta}{2(1+\beta)}\mathbf{E}\left[\langle r^k, \tilde{g}^k \rangle\right] \tag{21}$$

holds if the dynamics (2) reach stationarity. Both sides of (21) can be estimated by the history average during the training, thanks to ergodicity.

Unfortunately, evaluating this estimator requires 33% more training iterations than regular SGM due to the stochastic gradients used to compute the point $\tilde{x}^k$.

## C.2 Comparing stationary conditions

Figure 14 evaluates the two stationary conditions (5) and (6) by training an L2-regularized logistic regression model on MNIST and logging the estimators for both sides of each relation. The top row shows that even when the number of iterations grows large, there is still non-negligible error in Pflug's condition even though the function is strongly convex. Contrastingly, the statistics in Yaida's relation, shown in the bottom row, quickly become almost indistinguishable, as predicted by (6) and (4). Together with the difficulty of estimating its right-hand-side, this inaccuracy makes the Pflug condition unattractive for quantitative applications such as ours, which require a precise relationship to hold. However, the *qualitative* intuition given by such quadratic stationary formulae has proven useful (Mandt et al., 2017).

# D Additional SASA discussion

## D.1 The missing step to derive the stationary condition (6)

Assuming the existence of a stationary condition $\pi(d, x, g)$ for the SGM dynamics (2), Yaida (2018) showed

$$\mathbf{E}_\pi[\langle x, \nabla F(x) \rangle] = \frac{\alpha}{2}\frac{1+\beta}{1-\beta}\mathbf{E}_\pi[\langle d, d \rangle]. \tag{22}$$

Using history average to estimate the left hand side needs the full gradient of $F$, which is not available (or expensive to compute) during training. Instead, both Yaida (2018) and SASA use (6) in practice,

Figure 14: The two conditions (5) (top) and (6) (bottom) evaluated on a logistic regression model trained on MNIST, with $\alpha = 1.0$. Left two columns: iteration 0-100; Right two columns: iteration 10000-10100. In the statistics plots, the red and black curves (dark) are the running estimates of the left-hand and right-hand side of each condition, respectively. The light curves show the raw value of each estimator at each iteration. Even when the number of iterations is very large, the statistics suggested by (5) still do not match. On the other hand, the difference between the statistics in (6) quickly converges to zero.

i.e.,

$$\mathbf{E}_\pi[\langle x, g \rangle] = \frac{\alpha}{2} \frac{1+\beta}{1-\beta} \mathbf{E}_\pi[\langle d, d \rangle], \tag{23}$$

where the left hand side can be estimated with nearly no computational overhead. Here, we provide the missing step from (22) to (23).

By the law of total probability, we have

$$\mathbf{E}_\pi[\langle x, g \rangle] = \mathbf{E}_\pi \left[ \mathbf{E}_\pi[\langle x, g \rangle | x, d] \right] = \mathbf{E}_\pi \left[ \langle x, \mathbf{E}_\pi[g | x, d] \rangle \right].$$

We denote the time-independent transition kernel from $(x^k, d^k, g^k)$ to $(x^{k+1}, d^{k+1}, g^{k+1})$ in (2) as $\mathbf{T}$. Then since $\pi$ is the stationary distribution, we have the pushforward measure of $\pi$ under $\mathbf{T}$ is still $\pi$, i.e., $\mathbf{T}^\sharp \pi = \pi$. Then we have

$$\mathbf{E}_\pi[g | x, d] = \mathbf{E}_{\mathbf{T}^\sharp \pi}[g | x, d] \overset{(*)}{=} \mathbf{E}_{\mathbf{T}^\sharp \pi}[g(x) | x, d] = \nabla F(x),$$

where the definition of the transition kernel (2) is used in the step (*) and the unbiasedness of the stochastic gradient, see Eqn. (3), in the last step.

## D.2 Discussion on the multiple-test problem

Although SASA performs sequential hypothesis testing, it does not seem to suffer from the issue of inflated false discovery rate McDonald (2009). That is, we do not observe that the test fires earlier than it "should" in our numerical experiments. From Figure 5, we can see that the statistic $\bar{z}_N$ is either monotonically decreasing to 0 or first decreasing and then increasing to 0, leading to high positive correlation among the tests. This high correlation may prevent proportional inflated false discovery rates; see, e.g., Benjamini and Hochberg (1995); Blanchard and Roquain (2009); Lindquist and Mejia (2015).