[Reviews · NeurIPS 2019]

Reviewer 1



This paper studies how to test the stationarity of stochastic gradient with momentum using some advanced testing statistics that take the time correlations into accounts. Extensive experiments are run to demonstrate the advantage of the proposed method over existing approaches. Originality: The paper is based on extending a recent paper by Yaida. It does not seem that original to me but the authors do combine the condition by Yaida with some more advanced testing statistics in a new way. Overall I think the extension is quite natural, so the conceptual novelty is not that high. But to be fair, the paper still has presented some solid new results. Quality: I like the extensive experiments in this paper. It seems to me the quality of the experiments are reasonably good. On theoretical side, I like the intuitions in this paper: the time correlations have to be taken into accounts. However, it seems there is a gap between the assumptions in the theory and practice. The authors assume the gradient estimate is Markov in theory but in simulations they just run many different epochs. If I understand correctly, the data are reused when running different epochs. Then clearly there are time correlations between the reused data at different steps. It seems the Markov assumption is just too strong. If all the data are only used once and the gradient is sampled in an online manner, then the Markov assumption seems reasonable. But clearly the authors are not considering this case, since the online setting where the data is only used once just does not make sense for the overparameterized regions. Overall I don't think the proposed method in this paper has a solid theoretical foundation, although intuitively the Markov assumption is better than the IID assumption. The Markov assumption is not precise for the case where data is reused in multiple epochs. Another possibility is that fixing the data set and only considering the randomness in sampling. For this case, the Markov assumption seems fine but how to prove the time homogeneity here? Clarity: The paper is well written and quite clear. Significance: It is possible that the proposed testing statistics or some variants may be useful for practical problems although there are many other heuristics that can also be used to determine whether the SGD iterations have reached plateau or not. In addition, the authors' approach require setting another hyperparameter delta. This may introduce some extra difficulty in applying their approach for practical problems. Overall I think the overall significance is at the moderate level. =============================================== ============================================================ AFTER REBUTTAL: Thanks for the further explanations. The authors have addressed all concerns. I have increased my score.

Reviewer 2



The paper improves upon the main work from Yaida et. al by introducing a much more rigorous test for whether the chosen statistics indeed converge to zero. I think the authors introduction of the t-test and especially the improved estimators that take into account the temporal Markov Chain correlation is indeed great. The main text is very easy to follow and quite clear, with very clear idea and aim which the reader can follow. Several points of discussion: 1. I think a more important point though is that the authors should have plotted results with the learning rate adaptation of equation 9 for the comparison. From the variance tests and Fig.4 one can conclude that the newer test is better, but I think it is also important to measure to how much improvement in actual optimization that translates to? 2. There is an empirical analysis of the choice of the decaying constant (Fig. 3), however would be interesting to see similar plots for the confidence parameter (gamma in the paper) in the t-test interval construction. 3. On the comparison with ADAM it is claimed that it is "hand tuned". Given that SGM has decreasing learning rate schedule, there is no reason not to have run ADAM also with such cut-and-decay schedule and optimize it as well. 4. Any discussion on the higher order comparison rules from Yaida et. al would be beneficial. 5. It would be quite interesting to see if ADAM itself can benefit from SASA as well as how does the statistics of interest in the paper behave under ADAM in the first place. Figure 4 is really confusing as there is no legend or explanations in the figure title on what the different curves actually represent - I was expecting two lines for the two tests in eq.9 and 10 but there are clearly 5 (one figure has a single red curve)?

Reviewer 3



The paper proposes SASA, a method to drop the learning rate by a constant multiplicative factor when a certain criterion is met. The criterion is a statistical test, indicating stationarity of the Markov chain of the optimization parameters, which is performed once per epoch, and computed on easy to evaluate statistics of momentum SGD. The method is evaluated on three real world tasks and shows comparable performance to hand-tuned competitors. I. Originality Statistical tests for optimization are not new, in that regard the originality of the paper is low. However, I find the approach to test for stationary interesting and useful. II. Clarity The clarity of the paper is high. Indeed, while it is proposing a novel method, it contains parts similar to a technical report, which is refreshing to read. The authors also anticipate potential questions or concerns the reader might have and try to answer those in the paper already. Here are my concerns on clarity. i) The authors describe a “statistical test”, however, I am not sure if the exact test is defined somewhere in a concise way, e.g., what is the null-hypothesis? (Is it “no stationarity”?) ii) Furthermore, it is still not entirely clear to me what is at stake if the test triggers wrongly, or does not trigger (although it should). As I understand, the same type of test is performed at each epoch, which will make it somewhat likely that both those scenarios happen eventually. I could imagine for instance, the test not triggering, and the optimizer slowly diverging away from a stationary distribution again. On the other hand, non-stationarity seems to be the null-hypothesis, and not rejecting it should be the safe-place of the test. iii) In my opinion it would be beneficial to spend a bit more time on the interpretation of Yaida’s condition, as this is the condition of the proposed test. Looking closely, Yaida’s condition seems to say that the fraction of the squared norm of the search direction d_k, and the inner product of x_k and its gradient g_k is constant when SGM is in a stationary distribution. This, does not seem trivial at all to me, and I wonder if the authors have some intuition about why this relation holds (and what it means geometrically). iv) Additionally, from what I gather, the condition holds for “general functions F”; first, it is unclear to me what that means, and second, by assuming that there exists a stationary condition, one might implicitly also assume some smoothness on the function F that makes the stationarity possible. The condition, otherwise, seems very specific. v) On a practical note, it is unclear to me what the lowest possible number of samples N is that needs to be acquired to make the method work reliably. Especially for smaller datasets, this might be an issue, both because statistics lack after one epoch, and also because the Markov chain did not mix enough. vi) On another practical note, I am unsure, however, how applicable the proposed method is. For instance, I am unclear if the method is still applicable when the gradients g_k is a biased estimator of the gradient? Biased estimators seem to become popular lately, and the significance would increase or decrease depending on the applicability. [Post-Rebuttal: Thank you for your rebuttal. I increased my score, as I believe this is an interesting paper.]

[Author Response · NeurIPS 2019]

We thank the reviewers for their time, effort, and helpful feedback. We address individual comments below.

**Reviewer 1:** *Time homogeneity.* The *training* loss in all of our examples can be written in the form of lines 17-18. When each data batch $\xi$ is sampled uniformly with replacement, the time homogeneity follows from the form of the update dynamics, since $\alpha$ and $\beta$ in Eqn. (1) are constant while we collect samples and do statistical tests. The fact that the iterates of SGD form a homogeneous Markov chain is also used, for example, by Bach and Moulines (2013) and Dieuleveut et al. (2017). We will include these additional references and add a proof to the appendix. While we sample batches *without* replacement in the experiments, such practice is common in deep learning and is arguably a small gap.

*Statistics in loss space.* Consider equation (6) when the assumptions of Section 2.1 apply, i.e., $F(x)$ is a quadratic function and the additive noise to the gradient is independent of $x$. Then the left hand side of (6) is $\mathbf{E}_\pi[x^T A x]$, the mean value of the loss at stationarity, but the right-hand side is $\frac{\alpha}{2}\mathbf{tr}(\Sigma)$, the (scaled) trace of the noise covariance. In this case, we test whether the mean loss has converged to a constant, but we *also* have a different estimator for that constant. This test should be more sample-efficient than one that compares the mean loss to itself if the other estimator converges quickly. Our test can be considered as a more general version of testing whether the loss has reached a constant value.

*Hyperparameter $\delta$.* In our numerical experiments, we found that $\delta = 0.02$ worked well on all the examples we studied, and Appendix A contains a study of the sensitivity of the algorithm to changes of $\delta$ around this value. Intuitively, because the term that $\delta$ is multiplying is sensitive to the scale of the statistics, it is reasonable to expect roughly the same value of $\delta$ to work on different problems. In some sense, any method for testing stationarity must include a "slack term."

**Reviewer 2:** *Results with (9).* In addition to Figure 4, results for tuning using equation (9) are provided in Figures 12 and 13 in Appendix B. In short, (9) performs similarly to (10) on average, but has (potentially much) higher variance.

*Sensitivity to significance parameter.* The middle rows of Figures 6, 8, and 10 in Appendix A show the sensitivity of Algorithm 2 to changes in $\gamma$ around the default value of 0.2 on CIFAR-10, ImageNet, and MNIST, respectively.

*Hand-tuning Adam.* In the CIFAR-10 and ImageNet experiments, we used a hand-tuned "warmup phase" for Adam. The learning rates are not plotted here because they changed per parameter after the warmup phase. In the RNN example, the global learning rate of Adam is dropped based on the validation loss, and is simply missing from the bottom-right panel of Figure 2. We will add this global learning rate curve upon revision. It is similar to the one for SGM. Wilson et al. (2017, Section 4.2) observed that step-wise decay of Adam's global learning rate did not improve their results on CIFAR-10, so we only tuned the warmup phase for our image experiments.

*SASA for Adam.* Unfortunately, unlike SGM with fixed values of $\alpha$ and $\beta$, the dynamics of Adam depend heavily on time. Adam converges to a stationary point rather than to a stationary distribution with nonzero variance. This makes the SASA approach of testing for stationarity inapplicable to Adam without significant modification.

*Figure 4.* The five curves plotted are the performance across five independent runs. The $y$ axes are equalized throughout the figure, and in the (1,2) panel only one of the five curves is on the same scale as the others because of the variance of testing with (9). This is described in the main text, and we will update the Figure 4 caption to match the main text.

**Reviewer 3:** *Statistical testing.* The null hypothesis is that $|\mathbf{E}_\pi[\langle x, g\rangle] - \frac{\alpha}{2}\frac{1+\beta}{1-\beta}\mathbf{E}_\pi[\langle d, d\rangle]| \geq \Delta$; the alternative is that $|\mathbf{E}_\pi[\langle x, g\rangle] - \frac{\alpha}{2}\frac{1+\beta}{1-\beta}\mathbf{E}_\pi[\langle d, d\rangle]| < \Delta$. That is, the alternative is that the equation (6) holds up to a slack of $\Delta$. This is known as *equivalence testing* (Streiner, 2003). We have a relative threshold $\Delta = \delta\frac{\alpha}{2}\frac{1+\beta}{1-\beta}\mathbf{E}_\pi[\langle d, d\rangle]$ with the hyperparameter $\delta$ to make the threshold adaptive to the scale of the statistics. We can clarify the presentation of the test by including a brief overview of equivalence testing and by more explicitly stating the hypotheses. Intuitively, the null is "not stationary" and the alternative is "stationary," but with the caveats that we can only test equation (6) up to a slack term $\Delta$, and that equation (6) is merely necessary for stationarity, not sufficient.

*Interpreting Yaida's relation.* We gave some intuition on the condition in our "loss space" response to Reviewer 1 for quadratic $F$, which we can add after (6). By "general functions $F$" we mean any function of the form given in Section 1 such that the other assumptions (SGM→stationary distribution) apply. Understanding what these assumptions imply about $F$ seems to be quite challenging in the general nonconvex case, but the convergence of the statistics in Figures 5, 7, and 9 of the appendix suggests that they can hold in practice even for complicated, nonsmooth functions.

*Biased estimators.* It is unclear if the precise test used in this paper works when the gradient estimator is biased. Passing from Yaida's original formula (lines 121-122) to equation (6) requires unbiasedness. However, the general procedure of testing for stationarity still applies—the bias simply must be accounted for. We are pursuing some follow-up work to find more general stationary relations, but unbiased estimators remain the most common type in practice.

*Small datasets.* Note that the sample size $N$ is adaptive, but the *test frequency $M$* may need to be small for a small dataset. Approach (10) shines compared to (9) when there is large noise (due e.g. to a small $M$). When the variance of the statistics is high, not accounting for it can cause huge variance in the learning rate schedule, as in Figure 4.

[Meta-Review · NeurIPS 2019]

The paper proposes to automate the tuning of learning rate schedules in stochastic gradient methods, which is an important problem. In this regards, the authors propose a statistical test to determine when to decay the learning rate. The statistical test build upon a prior work with simple albeit useful extensions. Resulting statistical test is simple and can be deployed easily. There are some concerns regarding mismatch between theoretical assumptions made and the setup in practice. Nevertheless, empirically the learning rate schedule followed by decaying when the test is true seems to be almost competitive with hand-tuned methods. Thus, I am recommending an acceptance to NeurIPS. For the camera ready version, we would like the authors to re-evaluate their experimental results. Please make sure the standard train-test splits are used. Kindly repeat experiments a few times to get variability and error bars. Also ensure the comparisons made are in the correct ball park of known performance of the models (e.g. for wiki-text 2 perplexity seems a bit too high for the model used and accuracy of resnet-18-v1 from the cited paper on cifar10 seems to be too good).